# DNA Methylation Changes are Associated with the Programming of White Adipose Tissue Browning Features by Resveratrol and Nicotinamide Riboside Neonatal Supplementations in Mice

**DOI:** 10.3390/nu12020461

**Published:** 2020-02-12

**Authors:** Alba Serrano, Madhu Asnani-Kishnani, Charlene Couturier, Julien Astier, Andreu Palou, Jean-François Landrier, Joan Ribot, M. Luisa Bonet

**Affiliations:** 1Grup de Recerca Nutrigenòmica i Obesitat, Laboratori de Biologia Molecular, Nutrició i Biotecnologia (LBNB), Universitat de les Illes Balears, 07122 Plama, Spain; albaserranobengoechea@gmail.com (A.S.); madhuasnani@hotmail.com (M.A.-K.); andreu.palou@uib.es (A.P.); luisabonet@uib.es (M.L.B.); 2Center for CardioVascular and Nutrition research, UMR INSERM 1263, INRA 1260, Aix Marseille Université, 13385 Marseille, France; charlene.couturier@univ-amu.fr (C.C.); julien.astier@univ-amu.fr (J.A.); Jean-francois.LANDRIER@univ-amu.fr (J.-F.L.); 3Institut d’Investigació Sanitària Illes Balears (IdISBa), Palma de Mallorca, 07120 Palma, Spain; 4CIBER de Fisiopatología de la Obesidad y Nutrición (CIBEROBN), 28029 Madrid, Spain

**Keywords:** DNA methylation, metabolic programming, WAT beiging/browning, food bioactives, B vitamins, dietary polyphenols

## Abstract

Neonatal supplementation with resveratrol (RSV) or nicotinamide riboside (NR) programs in male mice brown adipocyte-like features in white adipose tissue (WAT browning) together with improved metabolism in adulthood. We tested the involvement in this programming of long-term epigenetic changes in two browning-related genes that are overexpressed in WAT of supplemented mice, *Slc27a1* and *Prdm16*. Suckling mice received orally the vehicle, RSV or NR from postnatal days 2-to-20. After weaning (d21) onto a chow diet, male mice were habituated to a normal-fat diet (NFD) starting d75, and split on d90 into continuation on the NFD or switching to a high-fat diet (HFD) until euthanization on d164. CpG methylation by bisulfite-sequencing was analyzed on inguinal WAT. Both treatments modified methylation marks in *Slc27a1* and *Prdm16* and the HFD-dependent dynamics of these marks in the adult WAT, with distinct and common effects. The treatments also affected gene expression of *de novo* DNA methylases in WAT of young animals (euthanized at d35 in independent experiments). Studies in 3T3-L1 adipocytes indicated the direct effects of RSV and NR on the DNA methylation machinery and favoring browning features. The results support epigenetic effects being involved in WAT programming by neonatal RSV or NR supplementation in male mice.

## 1. Introduction

Epigenetic changes modifying genome activity represent a prime candidate mechanism to explain the influences of early life nutrition on later metabolic phenotypes [1], including differences in the susceptibility to obesity and the metabolic syndrome [2]. Cytosine methylation at genomic CpG dinucleotides is an important epigenetic mechanism. Changes in DNA methylation may result in altered gene expression, leading to different phenotypes with potential increased or decreased disease risk. DNA methylation patterns are modified in response to dietary and other environmental stimuli [3], and relationships between early life nutrition, DNA methylation modifications, and the risk of metabolic disease have been described [1].

White adipose tissue (WAT) browning refers to the recruitment in white fat depots of brown-like (also called beige) adipocytes highly oxidative and thermogenic as compared to typical white adipocytes. WAT browning is an obesity-counteracting mechanism, and the browning potential of WAT is susceptible to nutritional programming [4]. The early postnatal period might be particularly important for this programming since it is a time-window of active WAT development [5]. We previously showed that mild supplementation during the suckling period with the polyphenol resveratrol (RSV, 3,5,4′-trihydroxy-trans-stilbene) or the B3 vitamin nicotinamide riboside (NR) promotes in mice the appearance of browning-related features in subcutaneous WAT in adulthood, together with better responses to a high-fat diet, selectively in the male progeny [6]. In good concordance, RSV supplementation of the maternal high-fat diet promotes brown and beige adipocyte development and prevents programmed obesity in the male offspring [7]. Mechanisms behind the programming of the adipose phenotype by neonatal RSV and NR supplementations likely include changes in WAT resident progenitor cells toward a greater commitment to beige (*versus* white) adipogenesis, as suggested by results in primary adipocyte cultures [8]. However, whether epigenetic changes affecting brown/beige adipocyte marker genes are involved in this programming has not been addressed.

*Slc27a1* (solute carrier family 27 member 1) and *Prdm16* (PR domain containing 16) are two browning-related genes that are induced in WAT of male mice treated neonatally with RSV or NR and derived primary cultures [6,8]. The protein product of *Slc27a1* is fatty acid transport protein 1 (FATP1), an integral membrane protein that mediates the cellular uptake of long-chain fatty acids and can channel the uptaken fatty acids toward oxidation in mitochondria [9]. SLC27A1/FATP1 is required for cold-induced thermogenesis in brown adipose tissue (BAT) [10], it is a proposed beige adipocyte transcriptional marker [11], and has been related to mitochondria function in white (3T3-L1) adipocytes [12]. Further, a non-hypothesis driven approach recently identified *Slc27a1* as a critical gene for the interactions of signaling pathways activated by exercise leading to subcutaneous WAT browning [13]. The protein product of *Prdm16*, meanwhile, is a key molecular determinant of brown/beige fat cell fate and a transcription coregulator that activates the thermogenic brown adipocyte gene program when expressed in cultured white preadipocytes or in WAT depots in vivo [14,15,16]. DNA demethylation at the *Prdm16* promoter is critical for BAT development during murine embryogenesis [17]. However, to our knowledge, studies investigating DNA methylation modifications of *Slc27a1* or *Prdm16* in connection to WAT browning or in response to dietary expositions (either in early or adult life) are lacking.

Here, we tested the hypothesis that long-term effects on DNA methylation marks in genes related to WAT browning, in particular, *Slc27a1* and *Prdm16*, are part of the programming of subcutaneous WAT elicited by early-life RSV and NR supplementations in male mice. We also studied the ability of these compounds to affect browning-related features and gene expression of the DNA methylation machinery in 3T3-L1 adipocytes, a widely used white adipocyte cell model.

## 2. Materials and Methods

### 2.1. Animal Experiments

DNA methylation analyses were conducted on inguinal WAT (iWAT) samples from previously phenotypically characterized cohorts of adult male mice that were supplemented during the suckling period with RSV or NR, and their non-supplemented controls. The study design is described in detail elsewhere [6]. The animal protocols were reviewed and approved by the Bioethical Committee of the University of the Balearic Islands (Ref. 3513, 26 March 2012). International standards for the use and care of laboratory animals were followed. Briefly, newborn NMRI mice in size-adjusted litters (12 pups/L) were randomly divided into three groups—control, RSV, and NR—which, from postnatal days 2 to 20, were given daily, with the aid of a pipette, an oral dose of 10–15 μL of vehicle (water), a RSV solution (Sigma-Aldrich, St. Louis, MI, USA, R5010), or a NR solution (Chemical Point, Oberhaching, Germany, CP1341-23-7-BULK), respectively. RSV was used at a dose of 2 mg/kg body weight, and NR, at a dose equivalent to ~15-fold the total vitamin B3 ingested daily from maternal milk [6]. Male animals were weaned on a standard chow diet (3.3 kcal/g, 8% calories from fat; Panlab, A04) on day 21 of age, and habituated to a purified normal-fat diet (NFD, 10% energy as fat; Research Diets, D12450J) from day 75 to day 90. Then, half the animals of each treatment group were switched to a purified high-fat diet (HFD, 45% energy as fat; Research Diets, D12451) while the other half remained on the NFD, making a total of 6 study groups (*n* = 5–6 animals per group, from 4–5 different litters). Animals were euthanized by decapitation at age 164 days, under fed conditions, within the first 2 h of the light cycle. Tissue biopsies were collected, snap-frozen in liquid nitrogen and stored at −80 °C until being processed. In two additional separate experiments, control and RSV-treated mice and control and NR-treated mice were sacrificed at age 35 days (*n* = 5–6 animals per group).

### 2.2. Tissue RNA Isolation and Gene Expression

Total RNA was extracted from iWAT samples using Tripure Reagent (Roche, Basel, Switzerland, 11667157001) according to the manufacturer’s instructions, followed by a sodium acetate/ethanol precipitation to purify nucleic acids. Isolated RNA was quantified using NanoDrop ND-1000 spectrophotometer (NanoDrop Technologies Inc., Wilmington, DE, USA) and its integrity confirmed by agarose gel electrophoresis. Reverse transcription, PCR amplification of selected cDNAs, and data analysis were as previously described [18]. Primer sequences used in the PCR reactions are available on request. Gene expression data were normalized against the ribosomal protein, large, P0 (*Rplp0*), and peptidylprolyl isomerase A (*Ppia*) as reference genes.

### 2.3. Tissue DNA Extraction and Bisulfite-Sequencing PCR

Genomic DNA was extracted with DNeasy Blood and Tissue kit (Qiagen, Hilden, Germany, 69504). CpG methylation of selected genomic zones was analyzed by direct sequencing as previously described [19,20]. Briefly, ~200 ng of DNA were bisulfite-converted using EZ DNA Methylation-Gold Kit (Zymo Research, Irvine, CA, USA, D5005), and 5 ng of this product was used for PCR amplification of each target genomic region of interest, using JumpStart™ REDTaq^®^ ReadyMix™ (Sigma-Aldrich, St. Louis, MI, USA P0982) and bisulfite-specific primers (obtained from Sigma-Aldrich) (Table 1). For each region, electrophoresis on a 2% agarose gel confirmed the integrity of the amplified DNA, which was then purified using the QIAquick PCR purification Kit (Qiagen, Hilden, Germany, 28104) and quantified with the NanoDrop ND-1000 spectrophotometer (NanoDrop Technologies Inc., Wilmington, DE, USA). Approximately 20 ng of DNA was used for direct sequencing using the BigDye^®^ Terminator v3.1 Cycle Sequencing Kit (Applied Biosystems, Foster City, CA, USA, 4337457) in an Applied Biosystems 2720 Thermal Cycler, and sequenced on a 3130 Genetic Analyser (Applied Biosystems). Sequences obtained were aligned to the original (unconverted) sequence using the BioEdit software for a quality control check (>80% alignment and >90% conversion rate). For each CpG site, the degree of C methylation was calculated using the formula: % methylated C = 100% × peak height C/(peak height C+peak height T) [21], using the BioEdit software.

Nucleotide sequences within the *Slc27a1* and *Prdm16* genes and upstream their transcription start sites (TSS) were obtained from EMBL-EBI and NCBI databases. Predicted transcription factor (TF) binding sites within the studied DNA regions were identified with Genomatix MatInspector software v3.7, using the “Adipose Tissue” filter.

### 2.4. Cell Experiments

Culture and adipose differentiation of murine clonal 3T3-L1 adipose cells were as previously described [22]. The experimental treatments were administered after 7 days of differentiation when ~80% of the cells were filled with intracellular lipid droplets. Total cellular RNA and DNA were extracted 24 h later for, respectively, gene expression analysis and global DNA methylation and cellular mitochondrial DNA content analyses. Separate experiments were conducted for RNA and DNA extraction.

#### 2.4.1. Cellular RNA Extraction and Gene Expression

Total RNA was extracted using TRIzol reagent (Thermo Fisher Scientific, Waltham, MA, USA, 15596026). cDNAs were produced from 1 μg of total RNA using random primers and reverse transcriptase. RT-qPCR analyses were performed on cDNAs using the Stratagene Mx3005P Real-Time PCR System (Thermo Fisher Scientific, Waltham, MA, USA,) as previously described [23]. Sequences of the corresponding primers used are available on request. Gene expression was quantified in duplicate. The ribosomal 18S rRNA was used as a housekeeping gene in the comparative cycle threshold method. The use of 18S as a reference gene was validated in previous experiments, which indicated that changes in 18S rRNA levels fit the best with the mean of the response of a series of candidate housekeeping genes assayed (including among others beta-actin, tubulin, *Gapdh, Hprt, Ppia*, and *Rplp0*) (results not shown).

#### 2.4.2. Cellular Mitochondrial DNA Content

Mitochondrial DNA (mtDNA) was estimated in duplicates using RT-qPCR, by measuring the threshold cycle ratio (ΔCt) of a mitochondrial-encoded gene (COX1) versus a nuclear-encoded gene (*Ppia*) as previously reported [24].

### 2.5. Statistical Analysis

Data are expressed as mean±SEM. Analyses were conducted separately by treatment (RSV or NR), with few exceptions. Two-way ANOVA was used to analyze treatment effects under NFD and HFD feeding conditions, followed by Fischer’s LSD post-hoc test whenever two-way ANOVA indicated an interactive effect between the treatment and the type of diet. The student’s *t*-test was used for single comparisons. Associations between *Slc27a1* and *Prdm16* mRNA levels and methylation status—global, and per all bisulfite-sequencing regions and individual CpG sites studied—were tested using Pearson’s correlation coefficients. One-way ANOVA followed by Fischer’s LSD post-hoc test was used for the analysis of dose-dependence effects in the adipocyte cell studies. The threshold of significance was set at *p* < 0.05. IBM SPSS Statistics for Windows, version 19.0 (IBM Corp.) was used for the analyses.

## 3. Results

### 3.1. Early Life Supplementation with RSV or NR Affected the Methylation Profile of Slc27a1 in iWAT of Adult Mice

Neonatal treatment with either RSV or NR results in an up-regulated expression of the beige adipocyte-selective marker *Slc27a1* in iWAT in adulthood in the male progeny [6]. *Slc27a1* up-regulation is especially evident under NFD conditions in the RSV mice (3.2-fold induction under NFD vs. 1.45-fold induction under HFD, relative to levels in corresponding controls), and under HFD conditions in the NR mice (2-fold induction vs. 1.15-fold induction under NFD) [6]. Furthermore, primary adipocytes derived from the stromal vascular fraction isolated from iWAT of young RSV-treated and NR-treated male mice also have *Slc27a1* gene expression up-regulated relative to corresponding primary adipocytes from control littermates [8]. These previous results, together with reports linking *Slc27a1* and its protein product FATP1 to thermogenesis, mitochondria function and WAT browning [10,11,12], made a priori *Slc27a1* an interesting gene to study its eventual epigenetic modification in response to the experimental treatments applied.

The basal promoter determinants of the murine *Slc27a1* gene lie 273 bp upstream three close transcription start sites (TSS) and DNA regulatory elements have been identified in more distal positions, up to −1353 bp [25,26]. We prospectively evaluated the methylation status of four *Slc27a1*-related bisulfite-sequencing regions (BSs): a distal promoter region (−1370 to −963, BS1), a more proximal promoter region (−472 to −117, BS2), and two overlapping intragenic regions, one spanning intron 1–2 and part of exon 2 (+135 to +527, BS3) and the other spanning part of exon 2 and of intron 2–3 (+504 to +904, BS4). The intragenic regions were analyzed in view of their enrichment in CpG dinucleotides (Figure 1A) and the fact that exons 1 and 2 are non-coding exons involved in alternative splicing events affecting *Slc27a1* expression [25]. In total, the methylation degree of 36 CpG sites located between positions –1370 and +904 relative to the predominant (according to [25]) *Slc27a1* TSS, which was taken as the reference point (+1), were successfully analyzed. Nine of these 36 CpG sites lie upstream the TSS, of which 5 are within BS1, and 4 within BS2; the other 27 CpGs analyzed lie in the intragenic BS3 and BS4 regions (Table 1 and Figure 1A). Mean methylation degree of all pooled BSs, of each BS, and of all specific CpG sites for which the methylation status changed significantly in response to the neonatal treatments, the type of diet, or their interaction according to two-way ANOVA are shown in Figure 1B, Figure 1C and Figure 1D, respectively.

Mean methylation degree near and within the *Slc27a1* gene considering all pooled BSs was significantly lower in the RSV and NR mice compared with the control mice (Figure 1B). More specifically, neonatal RSV treatment associated with hypomethylation of BS3, BS4, and specific CpG sites within these two regions (+448, +461, +547, +754, and +781), as well as with site-specific hypomethylation of CpGs -1184 and -1086 in BS1 under NFD (Figure 1C,D). Down-regulatory effects of neonatal NR treatment on DNA methylation, meanwhile, approached statistical significance for BS4 (*p* = 0.052, two-way ANOVA) and were significant for a number of specific CpG sites within BS3 and BS4 (namely CpGs +448, +461, +752, +754, and +792), as well as for mean BS1 methylation under NFD. Three CpG sites (+448, +461, and +754) were found to be hypomethylated in both RSV and NR mice compared with control mice. In all experimental groups, the methylation degree was maximal in the most upstream BS1 promoter region (~85% methylation), and minimal in the most downstream BS4 intragenic region (~25% methylation) (Figure 1C).

Our previous studies indicate that HFD feeding in adulthood does not affect *Slc27a1* mRNA levels in iWAT in control and RSV mice (although in the latter a trend to decreased expression levels was apparent) while it significantly increases these levels (by ~2-fold) in the NR mice [6]. Here, we found that HFD feeding in adulthood affects *Slc27a1* DNA methylation status in iWAT. Thus, compared to NFD-fed counterparts, control and RSV mice fed a HFD showed increased methylation of BS3 (global), and of CpG sites +301, +461, +554, and +689, and NR mice showed increased methylation of CpG site +754. Additionally, significant (or nearly significant) interactions of neonatal treatments with the type of diet affecting *Slc27a1* methylation were detected (TxD interactions). For mean BS1 methylation and site-specific methylation of CpGs −1184 and +801, these interactions were in the sense of the HFD up-regulating the methylation degree of these regions/sites in both groups of treated mice (which had lower corresponding methylation degrees than controls under NFD conditions) and down-regulating methylation in the control mice. In the RSV mice, additional TxD interactions in the same sense were observed for CpGs -1086 and +642. Whereas in the NR mice, TxD interactions in the opposite sense. i.e., HFD up-regulating methylation selectively in the control mice, and/or tending to down-regulate methylation selectively in the NR mice were apparent for BS2, BS3 and specific CpGs in these two regions, reaching significance for CpG +301 (Figure 1C,D and results not shown).

Associations between *Slc27a1* expression and methylation status in iWAT (considering the four BS regions together, per BS region, and per all individual CpG sites successfully analyzed) were assessed by linear Pearson’s correlation analyses. When all the animals were included in the analyses (*n* = 32–33), weak significant inverse correlations were observed between *Slc27a1* expression and mean methylation in all four BSs, BS3, and BS4 (Figure 2A). These inverse correlations were strengthened when just control and RSV mice were included (*n* = 20–21) (Figure 2B). At the individual CpG level, when all animals were included (*n* = 27–33), significant inverse correlations were detected involving CpG +448 (*r* = −0.404, *p* = 0.036), CpG +461 (*r* = −0.497, *p* = 0.007), and CpG +614 (*r* = −0.453, *p* = 0.008); again, when only control and RSV mice were included (*n* = 17−21), the aforementioned correlations were strengthened—CpG +448 (*r* = −0.644, *p* = 0.005); CpG +461 (*r* = −0.591, *p* = 0.005); CpG +614 (*r* = −0.603, *p* = 0.004)—and additional significant inverse correlations were evidenced involving CpG −1184 (*r* = −0.445, *p* = 0.049), CpG −1086 (*r* = −0.530, *p* = 0.016), and CpG +781 (*r* = −0.501, *p* = 0.024). On the contrary, when control and NR mice were analyzed together, no significant correlations between the transcriptional activity of *Slc27a1* and DNA methylation of any of these BS regions or CpG sites in iWAT were observed (Figure 2C and data not shown). Strikingly, a number of significant correlations between *Slc27a1* expression and site-specific methylation degree were detected for the NR mice (*n* = 10–12) that did not hold for the control or the RSV mice. These included strong direct correlations of *Slc27a1* expression with CpG −1184 (*r* = 0.779, *p* = 0.003), CpG −1159 (*r* = 0.802, *p* = 0.002), CpG +752 (*r* = 0.773, *p* = 0.003), and CpG +754 (*r* = 0.715, *p* = 0.009) mean methylation degrees, and an inverse correlation with CpG −319 (*r* = −0.709, *p* = 0.022) mean methylation degree. Altogether, the results suggest that distinct methylation-related regulatory mechanisms affect *Slc27a1* expression in iWAT in the NR mice, on the one hand, and the control and RSV mice, on the other.

The MatInspector software was used to identify putative binding sites for TFs related to adipose tissue biology that could be involved in *Slc27a1* transcription regulation. In BS1, putative binding sites for nuclear respiratory factor 1 (NRF1, starting at −1242), peroxisome proliferator-activated receptor gamma (PPARγ, starting at −1210), and carbohydrate response element-binding protein (CHREBP, starting at −1160) were identified. These three sites mapped close to CpG −1184, whose methylation degree was affected by TxD interactions for both neonatal treatments. In BS2, overlapping sites for PPARγ and the nuclear receptor TR4 were identified (starting at −469 and −468, respectively), which in fact are TFs reported to induce *Slc27a1* gene transcription in murine adipocytes through binding to these sites [26,27]. However, none of the CpGs analyzed within BS2 (of which the nearest to the PPARγ/TR4 putative site was CpG −436) showed significant methylation changes in response to the neonatal treatments, the type of diet or interactions thereof. In BS3, a putative binding site for CCAAT/enhancer-binding protein (starting at +204) mapped close to CpGs +184 and +198, and a putative binding site for MEL1 (MDS1/EVI1-like gene 1, which is an aliases for PRDM16) (starting at +323), close to CpG +321; again, methylation of these particular CpG sites was not affected by the neonatal treatments, the type of diet or interactions thereof. No adipose tissue-related TFs were detected by the MatInspector software in BS4.

### 3.2. Early Life Supplementation with RSV or NR Affected the Methylation Profile of Prdm16 in iWAT of Adult Mice

Early life supplementation with RSV or NR resulted in increased *Prdm16* mRNA levels in iWAT in adulthood in male mice, with a similar degree of induction for both treatments and after NFD or HFD feeding conditions (1.6–1.9-fold induction relative to expression levels in the corresponding controls) [6]. Moreover, *Prdm16* gene expression was also found induced in iWAT of young male mice neonatally treated with NR, and in the primary adipocyte cultures derived from it [8]. These results added to the role of PRDM16 as a brown adipocyte lineage-determining factor [14] and reported DNA methylation-dependent regulation of *Prdm16* gene expression [17], prompted us to assess for eventual effects of the experimental neonatal supplementations on the methylation profile of *Prdm16*.

*Prdm16* is enriched in CpG dinucleotides in the vicinity of the TSS, a characteristic of key developmental genes [28]. Bisulfite sequencing was used to analyze the DNA region between positions −339 and +170 around the *Prdm16* TSS (taken according to the EMBL-EBI database) (Figure 3A). This region contains 37 CpG sites, of which 18 could be successfully analyzed. Globally considered, the methylation degree across all CpG sites analyzed was low in iWAT (less than 3% under NFD) and, interestingly, it increased with HFD feeding in the control mice (up to ~75%) but not in the RSV- and the NR-treated mice, so that interactive effects with diet (TxD) were observed for both treatments (Figure 3B). Specifically, TxD interactions reached or approached statistical significance for both treatments for CpGs −35, −26, and +25, so that feeding a HFD in adulthood increased the methylation degree of these CpG sites in the control mice but not in the treated mice (Figure 3C). Additionally, HFD feeding associated with increased methylation of CpG −38 irrespective of the neonatal treatments, and with a trend toward increased methylation of CpG −178 in control and RSV mice (*p* = 0.056 for the diet effect, two-way ANOVA). Certain CpG sites in the *Prdm16* proximal promoter were found to be hypomethylated in the treated mice relative to controls irrespective of the type of diet, which were different in the NR mice (CpG −178) and the RSV mice (CpG −18) (Figure 3C).

Pearson’s correlation analysis was applied to assess for eventual links between *Prdm16* methylation status (global and of all individual CpG sites successfully analyzed) and *Prdm16* gene expression in iWAT. When all the animals were included in the analyses (*n* = 31–32), significant (or nearly) correlations were detected only for CpG −35 (*r* = −0.348, *p* = 0.055) and CpG +25 (*r* = −0.358, *p* = 0.044), both of them weak inverse correlations (Figure 4A and results not shown). When the correlation analysis was restricted to the HFD-fed groups (*n* = 15–16) inverse correlations involving CpG −35 and CpG +25 remained the only significant ones, and they were both strengthened (*r* = −0.524, *p* = 0.045 for CpG −35; *r* = −0.529, *p* = 0.035 for CpG +25) (Figure 4C). On the contrary, when the analyses were restricted to the NFD-fed groups (*n* = 16), these correlations were lost (Figure 4B), and none of the correlations tested attained statistical significance. These results suggest that the methylation degree of specific CpG dinucleotides (−35 and +25) may be particularly relevant as a determinant of *Prdm16* gene expression, especially under HFD feeding conditions.

*In silico* analysis with MatInspector indicated the presence in the *Prdm16* proximal promoter region analyzed of putative binding sites for TFs related to adipose tissue biology, in particular: a binding site for Sterol regulatory element-binding proteins 1 and 2 (SREBPs, starting at −92), three binding sites for NRF1 (starting at −261, −33, and +49, respectively), and a binding site for MEL1/PRDM16 itself (starting at −17). Interestingly, one of the NRF1 potential binding sites was near or overlapped multiple CpG sites (located from −35 to −18) whose methylation degree was affected by RSV or NR (alone or in interaction with diet). Further, the MEL1/PRDM16 putative binding site overlapped CpG −18, whose methylation degree was decreased in the RSV-treated mice relative to controls.

### 3.3. Early Life Supplementation with RSV or NR Affected the DNA Methylation Machinery in iWAT of Young Mice

We studied if the experimental neonatal supplementations could affect the expression of epigenetic modifiers in iWAT. To this end, mRNA levels of DNA methyltransferase (DNMT) and ten-eleven translocation (TET) enzymes—involved in the acquisition and the removal of the methylation mark, respectively [29,30]—were analyzed in iWAT of young (35-day-old) animals. RSV supplementation resulted in a down-regulated expression of genes for the two active *de novo* DNMTs, *Dnmt3a*, and *Dnmt3b*, whereas NR supplementation resulted in a down-regulated expression of *Dnmt3b* only (Figure 5). Neither treatment affected the gene expression of *Tet3*. Gene expression of the maintenance methylase *Dnmt1* was not detected in iWAT (data not shown).

### 3.4. RSV and NR Promoted Browning Features and Affected the DNA Methylation Machinery in 3T3-L1 Adipocytes

We conducted studies in mature clonal (3T3-L1) white adipocytes differentiated in culture to assess for direct effects of RSV and NR on adipose cells that could contribute to WAT remodeling as observed in vivo. 3T3-L1 adipocytes were used because they are a prototypical white adipocyte cell model rather than a canonical model of brown/beige adipocytes, yet they are capable of inducing brown adipocyte marker genes in response to certain pharmacological and nutritional factors (see for instance [24,31,32]). To this end, 3T3-L1 preadipocytes were stimulated to differentiate with a classical adipogenic hormonal cocktail and exposed once mature to RSV or NR for 24 h. Expression of *Slc27a1*, *Prdm16*, the thermogenic uncoupling protein 1 gene (*Ucp1*), and target genes related to mitochondria biogenesis and dynamics (*Tfam*, *Nrf2*, *Mnf2*) was analyzed first, together with surrogate measurements of mitochondria content. The results confirmed direct effects of both RSV and NR favoring certain browning features, as indicated by the induction of *Slc27a1*, *Tfam*, and *Nrf2* gene expression and the increase in the mitochondrial to nuclear DNA ratio (shown in Figure 6A,B for RSV treatment, and Figure 6D,E, for NR treatment). However, *Ucp1* gene expression in the 3T3-L1 adipocytes was unaffected by either treatment, while *Prdm16* expression was unaffected by RSV and decreased by NR exposure. Altogether, increased mitochondrial oxidative metabolism in the RSV- and the NR-treated mature 3T3-L1 adipocytes independent of the induction of *Ucp1* or *Prdm16* is suggested.

Expression of the same DNA methylation-related genes analyzed in iWAT of young mice (*Dnmt1*, *Dnmt3a*, *Dnmt3b*, and *Tet3*) was analyzed in the exposed adipocytes, together with that of Dnmt3-like (*Dnmt3l*) and of methionine adenosyltransferase II b (*Mat2b*). *Dnmt3l* encodes a catalytically inactive member of the DNMT3 family that interacts with and activates the functional *de novo* methylases 3A,B [33], although it may also elicit down-regulatory effects on genomic DNA methylation [34,35]. *Mat2b* encodes the regulatory subunit of methionine adenosyltransferase in the biosynthesis of the methyl donor S–adenosylmethionine. Both treatments affected the gene expression of the DNA methylation machinery in adipocytes yet in distinct manners. The effects of RSV treatment were restricted to the induction of *Dnmt1* and *Dnmt3l* (Figure 6C). NR treatment, meanwhile, did not affect *Dnmt1* expression (or that of *Dnmt3a*), repressed both *Dnmt3l* and *Dnmt3b* expression, and induced *Mat2b* and *Tet3* expression (Figure 6F). Thus, NR had wider effects than RSV on the expression of DNA methylation-related genes in 3T3-L1 adipocytes.

## 4. Discussion

Nutriepigenetics is particularly timely in the context of developmental programming, as early life nutritional exposures may condition epigenetic changes ultimately affecting gene expression and health in later life. We previously showed, in the same cohort of animals used in this work, that supplementation with RSV or NR during the suckling period favors in the male mice WAT browning features in subcutaneous adipose tissue (iWAT) in adulthood, including increased expression of *Slc27a1*, a thermogenesis-related fatty acid transporter, and *Prdm16*, a molecular determinant of brown/beige fat cell fate, together with better biometric and metabolic parameters and responses to an obesogenic diet [6]. In this work, results are presented that sustain the involvement of DNA methylation modifications in the long-term programming of increased *Slc27a1* and *Prdm16* expression in iWAT elicited by neonatal RSV and NR treatments. A growing number of studies implicate DNA methylation modifications in beneficial biological effects of RSV, yet most of these studies relate to cancer (reviewed in [36]) and only a few so far to metabolism [37,38], whereas the epigenetic activity of NR has not been previously addressed, to our knowledge.

Results presented suggest that neonatal RSV treatment-dependent hypomethylation near and within the *Slc27a1* gene may contribute to the increased *Slc27a1* mRNA expression levels shown in iWAT in adulthood. Thus, *Slc27a1* intragenic regions (BS3 and BS4) and specific CpGs in the *Slc27a1* distal promoter region (BS1) were found to be hypomethylated in iWAT of RSV mice compared with control mice, under NFD or both NFD and HFD. Moreover, inverse correlations between *Slc27a1* mRNA expression levels and the methylation degree across BS3, BS4, and of specific CpG sites (−1184, −1086, +448, +461, +614, +781) were found for pooled control and RSV mice. These results appear to be in good concordance with a general repressive effect of DNA methylation on gene expression, which is well established for central gene promoters [39]. Moreover, a DNA methylation-free region extending several hundred bases downstream of the TSS (as studied here for *Slc27a1*) may be a prerequisite for efficient transcription initiation [40]; whereas DNA methylation in the gene body downstream might reduce or enhance transcription elongation efficiency [39,41].

Results regarding the impact of NR treatment on *Slc27a1* are of more complex interpretation. NR mice also had decreased *Slc27a1*-related methylation marks in iWAT compared to controls. However, correlations between *Slc27a1* gene expression and methylation degrees observed for pooled control and RSV mice disappeared for pooled control and NR mice. Moreover, results herein suggest that NR supplementation turns on specific *Slc27a1* modulatory mechanisms not operating or of less importance in the other two groups. First, NR mice had distinct *Slc27a1* methylation responses to the HFD compared to both control and RSV mice, as they distinctly lacked HFD-induced hypermethylation or even showed trends to HFD-induced hypomethylation of BS2 and certain CpG sites in BS3 and BS4. Although these effects were subtle and most of them statistically non-significant due to the low number of animals, it is remarkable that a HFD-dependent increase in iWAT *Slc27a1* expression was present in the NR mice only [6]. Second, the NR mice showed a distinct HFD-induced increase in the overall methylation degree of *Slc27a1* BS1. HFD feeding often results in hyperinsulinemia [42], and BS1 covers a cis-acting element (between −1347 and −1353) reported to mediate the negative effects of insulin on *Slc27a1* gene transcription in adipocytes [25]. It could be, therefore, that increased global BS1 methylation in the NR mice under HFD interferes with this repressive mechanism, thus contributing to sustaining *Slc27a1* transcription, and somehow to defend against negative effects of HFD-induced hyperinsulinemia. It is noteworthy in this context that, among the NR mice (but not the other treatment groups), greater *Slc27a1* gene expression correlated with greater methylation at certain CpG sites within BS1 (−1184 and −1159).

The other gene in our focus, *Prdm16,* is expressed in BAT and selectively in subcutaneous WAT relative to other white fat depots in mice [15]. The low methylation degree of the *Prdm16* proximal promoter found here in subcutaneous (inguinal) WAT may fit with this pattern of expression, considering the well-established association between promoter methylation and transcription silencing [39]. Interestingly, RSV and NR neonatal treatments promoted further hypomethylation of distinct CpG sites in the *Prdm16* proximal promoter irrespective of the type of diet and blunted the HFD-induced hypermethylation of other CpG sites in the *Prdm16* promoter observed in iWAT of control mice, all of which may contribute to increased *Prdm16* expression. The CpG sites whose methylation was affected by treatments already under NFD conditions were different in the RSV and the NR mice (CpG −18 and CpG −178, respectively), whereas the CpG sites that became relevant under HFD conditions were the same for both treatments (mainly CpGs −35 and +25). To be noted, the BS region studied encompasses *Prdm16* promoter-proximal parts that are actively demethylated during brown adipogenesis, paralleling increased *Prdm16* transcription [17]. Epigenetic regulation of *Prdm16* through DNA methylation modification has also been demonstrated in peripheral blood of adolescents exposed to maternal diabetes in utero [43], the placenta of mothers with gestational diabetes [44], and in connection to cancer [45].

DNA methylation may alter the affinity of DNA-binding proteins including TFs for their cognate sites in DNA, thereby affecting transcription and other aspects of genome function [46,47]. Interestingly, putative binding sites for TFs known to be relevant in the context of adipose tissue and energy metabolism overlapped or located near CpG sites affected by the experimental neonatal interventions applied. In particular, putative binding sites for NRF1—a TF related to mitochondria biogenesis and dynamics [48]—were identified in the *Slc27a1* distal promoter and *Prdm16* proximal promoter regions analyzed, close to CpGs that were found to be hypomethylated in the treated animals, under NFD or both NFD and HFD. Importantly, *Slc27a1* and *Prdm16* are both putative target genes of NRF1 according to ChIP-seq results in the ENCODE Transcription Factor Targets data set [49,50]. Moreover, RSV [51,52,53] and NR [54] both have been shown to induce NRF1 expression in experimental models as part of their stimulatory effects on mitochondrogenesis mediated by the activation of the SIRT1-PGC1α axis. Altogether, it is suggested that NRF1 may play a role in linking the observed effects of the neonatal treatments applied to methylation and transcription of both studied genes. For *Slc27a1*, a possible involvement of PPARγ is also suggested; transcription of *Slc27a1* is induced following PPARγ activation in adipocytes [26], and MatInspector identified putative PPARγ binding sites not only in the more proximal promoter region as expected [26], but also in the *Slc27a1* distal promoter region near a CpG (−1184) found to be hypomethylated in the treated mice relative to controls under NFD. In the case of *Prdm16*, besides control by NRF1, our results are suggestive of an auto-stimulatory transcriptional loop promoted by neonatal RSV treatment through the hypomethylation of specific CpG sites (notably CpG −18) that overlapped a putative PRDM16 binding site in the *Prdm16* proximal promoter (note that, although best known as a transcriptional comodulator, PRDM16 can also bind directly to DNA through its zinc-finger domains [55]).

How can the neonatal treatments applied affect DNA methylation marks in adipose tissue in adulthood, as well as the HFD-dependent dynamics of these marks? A possibility could involve the NAD^+^-dependent protein deacetylase sirtuin 1 (SIRT1). RSV and NR are both described as activators of SIRT1 (though through different mechanisms) [51,56,57], and SIRT1 has epigenetic activity, as it impacts global and gene-specific DNA methylation [58,59,60] and interacts with and deacetylates components of the DNA methylation machinery [60,61,62]. Still, RSV and NR have additional, non-overlapping biological targets for interaction besides SIRT1, and in fact, it is to be noted that effects of neonatal RSV and NR treatments observed in our previous studies [6,8] and in this work are similar, but not identical.

RSV and NR neonatal treatments mostly had down-regulatory effects on the methylation degree of studied genes in iWAT of adult male mice and, interestingly, they both exerted down-regulatory effects on gene expression of main *de novo* DNA methylases in iWAT of young mice. This suggests one mechanism that could explain, through persistent epigenetic changes, some of the treatment-dependent DNA methylation modifications observed, although how CpG site-specific hypomethylation would be achieved remains unclear. Additionally, the question remains how exposure to a (nutritional) factor early in life can condition the response of methylation marks to a stimulus faced later in life, as observed here in both groups of treated mice upon challenging with a HFD in adulthood. Besides programming effects at the peripheral tissue level, programming effects at the central (brain) level are likely to be involved in this type of epigenetic memory [1]. Notwithstanding model limitations, cell studies in this work indicate that RSV and NR may have direct effects in adipocytes impacting beige features and gene expression of components of the DNA methylation machinery. The effects on the methylation machinery in 3T3-L1 adipocytes were substantially different for the two compounds and, in the case of RSV, different from corresponding effects in iWAT of young treated mice. In particular, down-regulation of the *de novo* DNA methylases *Dnmt3a* and *Dnmt3b* found in iWAT of RSV mice was lacking in the RSV-exposed adipocytes, suggesting that indirect mechanisms are in place in vivo. Further studies are needed to answer mechanistic questions and to probe working hypotheses arising from this work.

## 5. Conclusions

In conclusion, this work shows that DNA methylation modifications may contribute to the long-term programming of an up-regulated expression of browning-related genes *Slc27a1* or *Prdm16* in subcutaneous WAT brought about by moderate supplementation of RSV or NR during the suckling period in male mice. To our knowledge, this is the first study to report changes in the DNA methylation status of *Slc27a1* or *Prdm16* in connection to WAT browning and epigenetic modification in connection to a dietary NR intervention. Together with our previous studies [6,8], results suggest RSV and NR have potential as active ingredients in early life strategies of obesity prevention in the long-term, particularly in male offspring.

## Figures and Tables

**Figure 1 nutrients-12-00461-f001:**
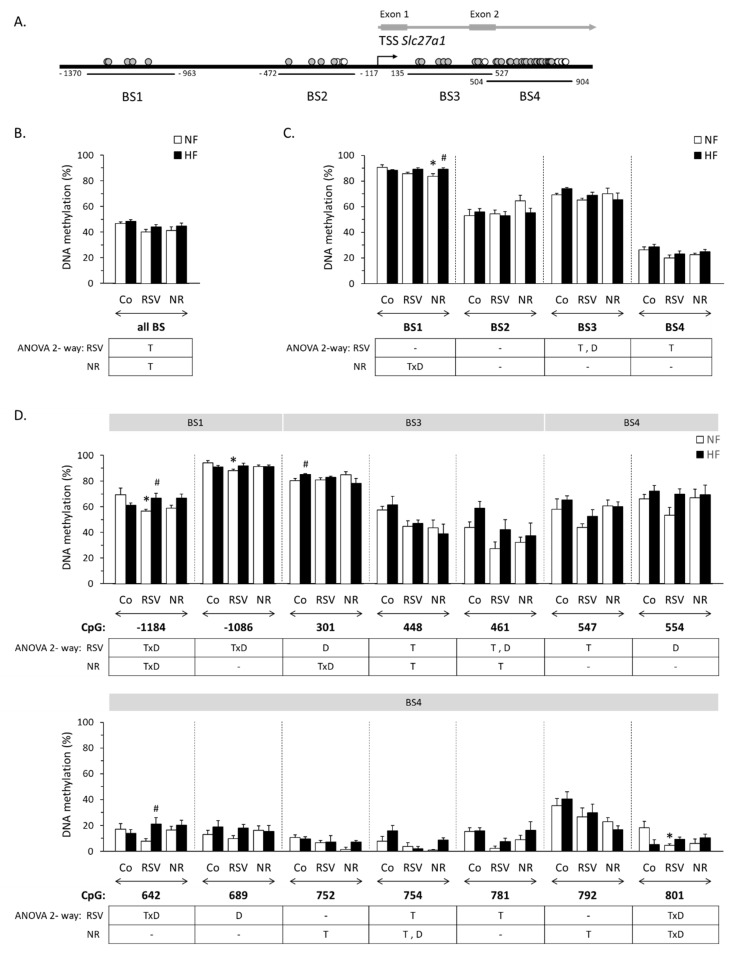
(**A**): Localization of the analyzed bisulfite-sequencing regions (BS) upstream and downstream the mouse *Slc27a1* gene transcription start site (TSS, +1). Filled circles represent individual CpG sites whose methylation status could be successfully analyzed; empty circles represent CpG sites that could not be analyzed. (**B**,**C**): Mean methylation degree in all four *Slc27a1* BS regions (**B**) and per BS region (**C**) in inguinal WAT of male mice neonatally treated with resveratrol (RSV) or nicotinamide riboside (NR) and submitted in adulthood to a HF/NF (high-fat/normal-fat) challenge, and corresponding control mice (Co, vehicle-treated). (**D**): Mean methylation degree in the individual CpG sites in *Slc27a1* BS1 to BS4 regions (as indicated) for which there were significant changes in methylation degree in response to the neonatal treatments, the type of diet or their interaction according to ANOVA 2-way analysis; other individual CpG sites analyzed are not represented. Data: mean ± SEM of 5-6 animals per group. Statistics (*p* < 0.05): T treatment, D diet, TxD treatment x diet interaction (ANOVA 2-way); * RSV or NR vs. control, ^#^ HF vs. NF (*t*-test).

**Figure 2 nutrients-12-00461-f002:**
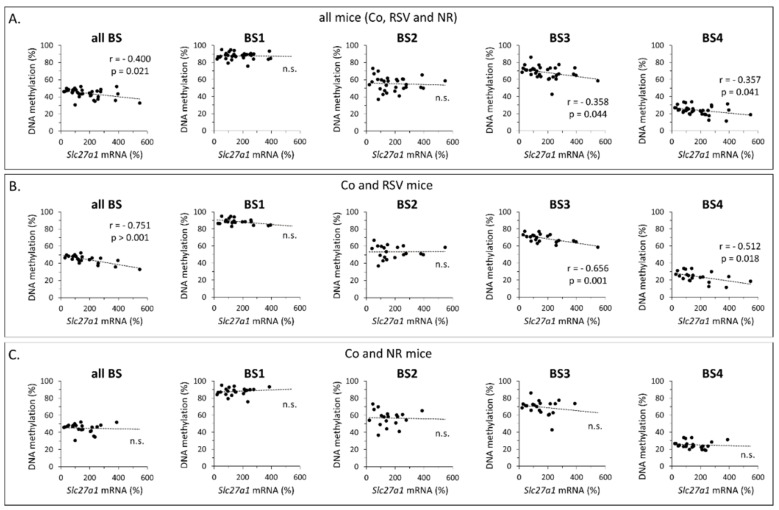
Association between *Slc27a1* expression (mRNA levels) and *Slc27a1* methylation status in inguinal WAT of male mice neonatally treated with resveratrol (RSV) or nicotinamide riboside (NR) and submitted in adulthood to a high-fat/normal-fat challenge, and corresponding control mice (Co, vehicle-treated). Pearson’s correlation analyses between *Slc27a1* expression levels and methylation status of all four pooled bisulfite-sequencing regions (BS) analyzed and per BS region are shown for all animals in the study. Co, RSV and NR (**A**), Co and RSV mice (**B**), and Co and NR mice (**C**). For each analysis, r and p values are indicated.

**Figure 3 nutrients-12-00461-f003:**
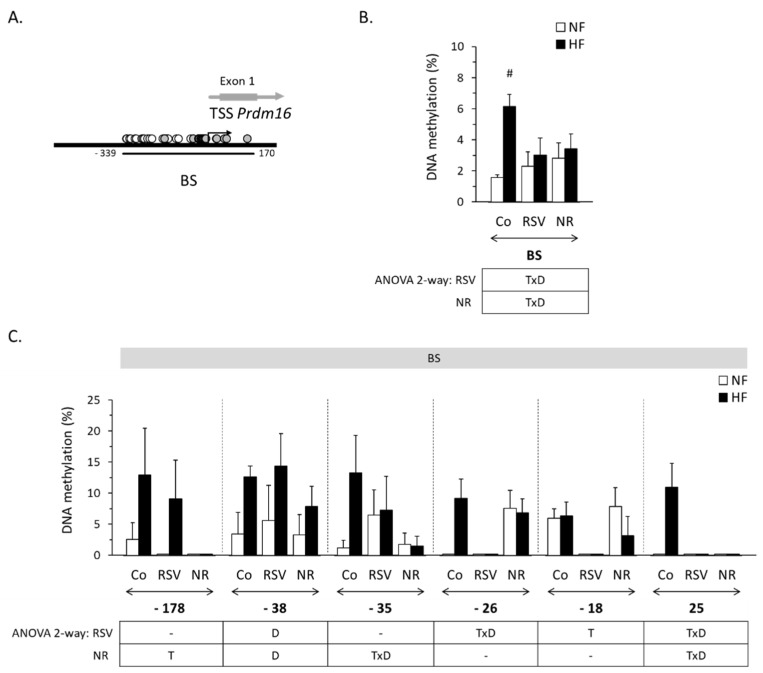
(**A**): Localization of the analyzed bisulfite-sequencing region (BS) surrounding the mouse *Prdm16* gene transcription start site (TSS, +1). Filled circles represent individual CpG sites whose methylation status could be successfully analyzed; empty circles represent CpG sites that could not be analyzed. (**B**): Mean methylation degree in the *Prdm16* BS region in inguinal WAT of male mice neonatally treated with resveratrol (RSV) or nicotinamide riboside (NR) and submitted in adulthood to a HF/NF (high-fat/normal-fat) challenge, and corresponding control mice (Co, vehicle-treated). (**C**): Mean methylation degree in all individual CpG sites in the *Prdm16* BS region analyzed for which there were significant changes in methylation degree in response to the neonatal treatments, the type of diet or their interaction according to ANOVA 2-way analysis; other individual CpGs in the BS are not represented. Data in B and C: mean ± SEM of 5-6 animals per group. Statistics (*p* < 0.05): T treatment, D diet, TxD treatment x diet interaction (ANOVA 2-way); * RSV or NR vs. control, ^#^ HF vs. NF (*t*-test).

**Figure 4 nutrients-12-00461-f004:**
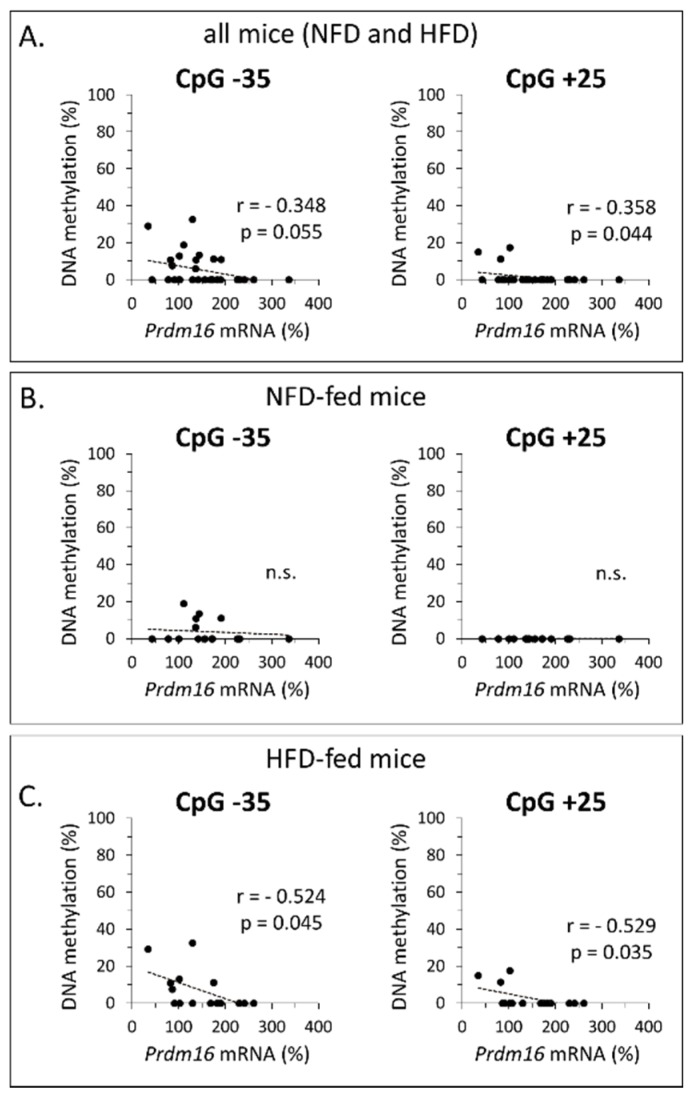
Pearson’s correlation analyses between *Prdm16* mRNA expression levels in inguinal WAT and site-specific *Prdm16* promoter methylation at CpG −35 and +25. Analyses were performed using data from male mice neonatally treated with resveratrol or nicotinamide riboside and submitted in adulthood to a high-fat/normal-fat diet (HFD/NFD) challenge and their corresponding vehicle-treated controls. Correlations are shown for: all animals in the study (**A**), the NFD-fed animals only (**B**), and the HFD-fed animals only (**C**). For each analysis, r and p values are indicated. No significant correlations between methylation degree and gene expression levels were observed for any other CpG in the *Prdm16* promoter analyzed.

**Figure 5 nutrients-12-00461-f005:**
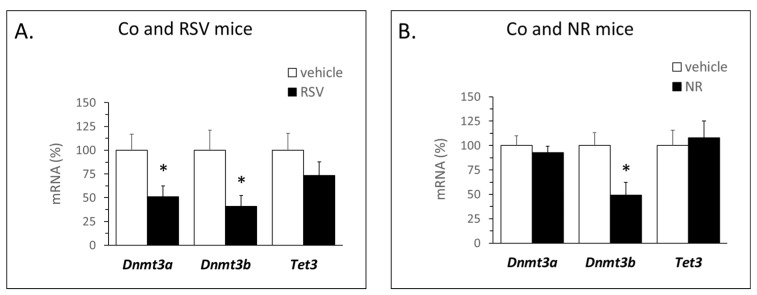
mRNA levels of DNA methylation-related genes *Dnmt3a*, *Dnmt3b*, and *Tet3* in inguinal WAT of young (35-day-old) male mice treated orally with vehicle (control), resveratrol (RSV, **A**) or nicotinamide riboside (NR, **B**) during the suckling period. Data: mean ± SEM of *n* = 5–6 animals per group (mean value in controls set at 100%). Statistics: * *p* < 0.05, effect of treatment, Student’s *t*-test.

**Figure 6 nutrients-12-00461-f006:**
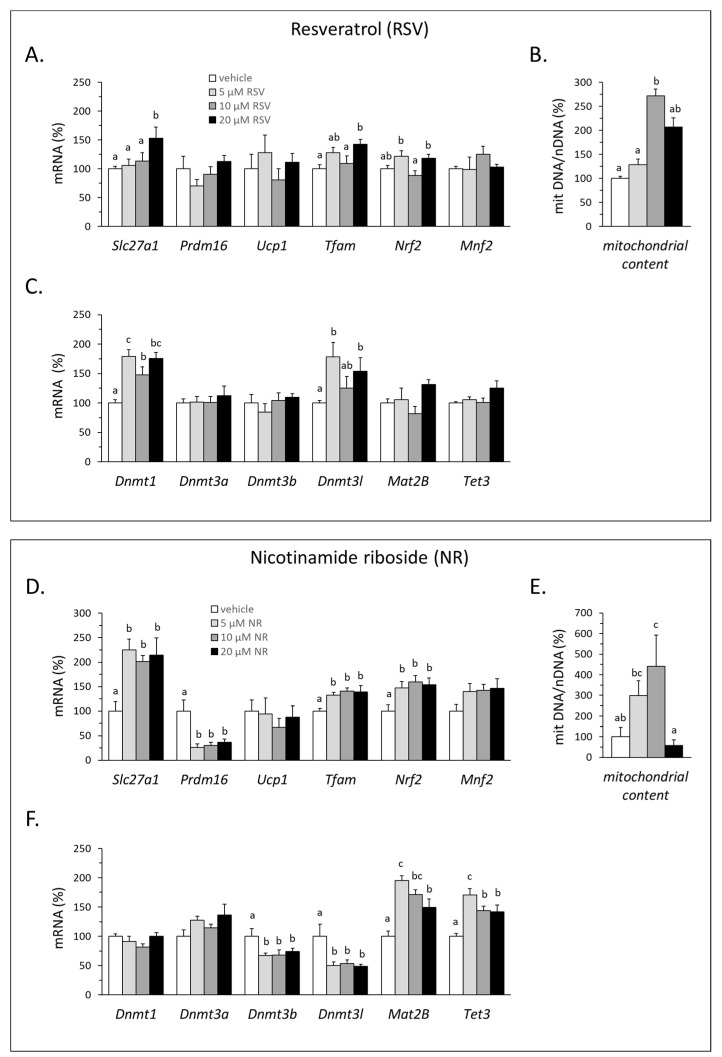
mRNA levels of indicated genes related to adipocyte browning (**A**,**D**), mitochondrial DNA to nuclear DNA ratio as an indicator of mitochondrial content (**B**,**E**), and mRNA levels of indicated genes related to the methylation machinery (**C**,**F**) in differentiated 3T3-L1 adipocytes exposed to the indicated agents and doses for 24 h. Data are expressed as mean ± SEM of *n* = 4 cultures per dose. Statistics: values not sharing a common letter are different with *p* < 0.05 according to ANOVA one-way, followed by LSD test.

**Table 1 nutrients-12-00461-t001:** Bisulfite-specific primers for analyzed DNA regions of *Slc27a1* and *Prdm16* genes.

Gene	Region	Gene Forward (f) and Reverse (r) Primers(5′→3′)	AmpliconSize (bp)	Total Numberof CpGs	Number of CpGsAnalyzed
*Slc27a1*	BS1	f: TGTTTTTATGGTGAGGAGAGGAAAATATGT	392	5	5
		r: TTACCCAAAAAACAAAAAATCCTAAAATCC			
	BS2	f: GTGGGGTAAAGGGTATAGGAGATGTTTTAG	356	7	4
		r: CCTTCCCACAACTCTCCTTAAAAAAAA			
	BS3	f: TTTTGATAGTAAGGGTGGGGGTATTTTAGTA	39	9	7
		r: ACCTAATCCAACTTATCCTAAATCCAAACC			
	BS4	f: GATTTAGGATAAGTTGGATTAGGTAAGTTT	401	25	20
		r: ACAATCACTATTCACAAAAAAACCC			
*Prdm16*	BS	f: ATTTAAGGAAGTTGTGTAGAAATTT	508	37	18
		r: CCTTAAATCACATAATATCAACTCA

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
