# Peer review of "DNA Methylation Changes are Associated with the Programming of White Adipose Tissue Browning Features by Resveratrol and Nicotinamide Riboside Neonatal Supplementations in Mice"

_nutrients, 2020, doi:10.3390/nu12020461_

Round 1

Reviewer 1 Report

Thanks to authors for this interesting and elegant study. The in vivo part of this work is robust, well-conducted and support completely the discussion and conclusion of the authors. Nevertheless, I was deeply disturbing by the inconsistency of in vitro part, too preliminary and not really well-conducted.
Indeed:
- 3T3-L1 cells are, for my point of view, an acceptable model for white adipocyte analysis, despite numerous limitations, but one of the worst models for brown/beige adipocytes. This is due to their low capacity to be convert into thermogenic adipocytes, the very low functional impact of this limited conversion and the discrepancies for several gene expression found in true thermogenic adipocytes. Moreover, analysis of methylation/epigenetic events in a clonal cell line is poorly relevant. Only, primary adipocytes derived from SVF fraction of white adipose tissue can be considered as a good model, especially for methylation analysis.
- the choose of 18S rRNA as house keeping gene is not acceptable, due to its quantity difference compare to others mRNA leading to normalization errors, differently to house keeping genes from in vivo part (Rplp0 and Ppia) which are robust.
- the lack of functional analysis and of specific methylation analysis make this part too preliminary.
- the inconsistency between gene expression, 5m-C results and expected effect.
Due to all these reasons and since this in vitro part seems me useless to support the conclusion, I propose to the authors to exclude this small in vitro part from this work.
Additional comments:
- Correlation between prdm16 expression and methylation level should be displayed in a figure.
- To complete figure 4, what is the expression DNA methylation related genes in HFD conditions with or without treatment?

Author Response

Manuscript ID: nutrients-709360

Title: DNA methylation changes are associated with the programming of white adipose tissue browning features by resveratrol and nicotinamide riboside neonatal supplementations in miceCover letter

We are sending you the revised version of this Ms, to be considered for publication in this revised form in Nutrients, and in particular, within the Special Issue "Nutrition and Epigenetics" (Editors: Giuseppe Passarino, Alberto Montesanto, Dina Bellizzi) (https://www.mdpi.com/journal/nutrients/special_issues/Nutrients_Epigenetics).

The original Ms was well evaluated and required minor revision only according to the two Reviewers. In preparing this new version, we have taken into account all the constructive criticisms and suggestions raised by the Reviewers, to whom we thank very much for their time and comments, as we believe the Ms improves by incorporating them.

A detailed letter explaining changes introduced is included. Changes introduced in the Ms have been highlighted in yellow.

Hoping that in its revised form you will find the manuscript suitable for publication, and looking forward to hearing from you soon.

Answers to the referees

Comments received are in blue and cursive; answers are in black.

 Reviewer 1:

Thanks to authors for this interesting and elegant study. The in vivo part of this work is robust, well-conducted and support completely the discussion and conclusion of the authors. Nevertheless, I was deeply disturbing by the inconsistency of in vitro part, too preliminary and not really well-conducted.
Indeed:
- 3T3-L1 cells are, for my point of view, an acceptable model for white adipocyte analysis, despite numerous limitations, but one of the worst models for brown/beige adipocytes. This is due to their low capacity to be convert into thermogenic adipocytes, the very low functional impact of this limited conversion and the discrepancies for several gene expression found in true thermogenic adipocytes. Moreover, analysis of methylation/epigenetic events in a clonal cell line is poorly relevant. Only, primary adipocytes derived from SVF fraction of white adipose tissue can be considered as a good model, especially for methylation analysis.

- the choose of 18S rRNA as house keeping gene is not acceptable, due to its quantity difference compare to others mRNA leading to normalization errors, differently to house keeping genes from in vivo part (Rplp0 and Ppia) which are robust.
- the lack of functional analysis and of specific methylation analysis make this part too preliminary.
- the inconsistency between gene expression, 5m-C results and expected effect.
Due to all these reasons and since this in vitro part seems me useless to support the conclusion, I propose to the authors to exclude this small in vitro part from this work.

We appreciate Reviewer 1 overall liked the work and considered it interesting, and we thank very much his/her congratulating words.

Regarding the use of 3T3-L1 adipocytes, we agree this model has limitations, and that, especially for DNA methylation analyses, primary adipocytes derived from SVF fraction of WAT is a better model (though in our hands difficult to establish from mature mice). We also agree that the level of 5m-C may be not relevant, since it only represents global methylation levels, and therefore these data are now excluded. Nevertheless, the 3T3-L1 model is widely used for screening and mechanistic studies, and we believe it is an appropriate model to answer the question whether tested compounds (RSV and NR) may have a direct impact on browning features and the methylation machinery in mature adipose cells. To this end, the use of 3T3-L1 adipocytes may even have advantages, since these cells are prototypical white adipocytes (rather than a model of brown/beige adipocytes), yet they are capable of inducing brown adipocyte marker genes in response to certain pharmacological and nutritional agents, as showed in multiple papers in the literature using this model. Assays in this context using this model would thus represent a sort of strict test.

Regarding the use of 18S rRNA as housekeeping gene, we are aware of the controversy, with many papers published on this topic, some pro, some con. Importantly, 18S has been validated in our experimental 3T3-L1 model as a good internal control, compared to many other housekeeping genes that we tested by the past (beta actin, tubulin, GAPDH, HPRT, cyclophilin, 36B4, etc.). When making a mean of the response of all these housekeeping genes, we have seen that 18S was the one which fit the best with the mean of all these references. This is now stated in Materials and Methods.

All in all, consequently to the Reviewer 1 suggestions, the following changes have been introduced in the Ms:

- results on global 5m-C methylation in 3T3-L1 cells have been excluded from the text and former Figure 5 (now Figure 6);

- a sentence on the suitability of 18 S rRNA as the reference gene in the 3T3-L1 gene expression analyses has been included in section 2.4.1. Cellular RNA extraction and gene expression (lines 157-162);

- a comment on the rationale of using 3T3-L1 adipocytes has been added at the beginning of section 3.4. on 3T3-L1 results, together with three illustrative references (references 24, 31 and 32 of the revised version; the former a previous collaborative article among authors that was already cited in the original version, and the other two new additions);

- that the 3T3-L1 model has limitations is acknowledged in the discussion of results obtained in this model (line 526).

Additional comments:
- Correlation between prdm16 expression and methylation level should be displayed in a figure.

A figure showing these results has been added (current Figure 4).

- To complete figure 4, what is the expression DNA methylation related genes in HFD conditions with or without treatment?

Expression of DNA methylation related genes was assayed in young, 35-day-old, animals, shortly after completing the neonatal treatments, in an attempt to find biological clues in the young animals that could relate to later methylation phenotypes (discussed at the beginning of the penultimate paragraph of the Discussion, lines 516-521). Since all animals were on the same chow diet from weaning to day 35, there are not HFD groups at this age.

Reviewer 2 Report

The current manuscript by Serrano et al., studied epigenetic programming pertaining to DNA methylation changes in white adipose tissue browning by resveratrol (RSV) and nicotinamide riboside (NR). The study focus on the involvement of Scl27a1 and Prdm6 genes in programming long-term epigenetic changes. This will give an insight on the potential use of RSV and NR as active ingredients in early life strategies of obesity in long term. The experiments appear carefully designed with the rationalized techniques. Results represented systematically and as easily comprehensible using appropriate statistical methods.  

I have few basic concerns regarding the transcriptional aspects, which can be included in introduction or discussed by citing proper literatures; What are the transcription factors responsible for the induction of Scl27a1 and Prdm6? Are both transcribed by same transcription factor or different?

Why the above question makes its important point because, RSV and NR may have direct role in inhibiting these transcription factors which may lead to positive\negative feed back on the Scl27a1 and Prdm6. This needs to be discussed by siting previously published work of direct effects of RSV and NR on transcription factors.

Author Response

Manuscript ID: nutrients-709360

Title: DNA methylation changes are associated with the programming of white adipose tissue browning features by resveratrol and nicotinamide riboside neonatal supplementations in mice

 Cover letter

We are sending you the revised version of this Ms, to be considered for publication in this revised form in Nutrients, and in particular, within the Special Issue "Nutrition and Epigenetics" (Editors: Giuseppe Passarino, Alberto Montesanto, Dina Bellizzi) (https://www.mdpi.com/journal/nutrients/special_issues/Nutrients_Epigenetics).

The original Ms was well evaluated and required minor revision only according to the two Reviewers. In preparing this new version, we have taken into account all the constructive criticisms and suggestions raised by the Reviewers, to whom we thank very much for their time and comments, as we believe the Ms improves by incorporating them.

A detailed letter explaining changes introduced is included. Changes introduced in the Ms have been highlighted in yellow.

Hoping that in its revised form you will find the manuscript suitable for publication, and looking forward to hearing from you soon.

Answers to the referees

Comments received are in blue and cursive; answers are in black.

 Reviewer 2:

The current manuscript by Serrano et al., studied epigenetic programming pertaining to DNA methylation changes in white adipose tissue browning by resveratrol (RSV) and nicotinamide riboside (NR). The study focus on the involvement of Scl27a1 and Prdm6 genes in programming long-term epigenetic changes. This will give an insight on the potential use of RSV and NR as active ingredients in early life strategies of obesity in long term. The experiments appear carefully designed with the rationalized techniques. Results represented systematically and as easily comprehensible using appropriate statistical methods.

We appreciate Reviewer 2 overall liked the work and considered it interesting, and we thank him/her very much for positive comments.

I have few basic concerns regarding the transcriptional aspects, which can be included in introduction or discussed by citing proper literatures; What are the transcription factors responsible for the induction of Scl27a1 and Prdm6? Are both transcribed by same transcription factor or different?

Why the above question makes its important point because, RSV and NR may have direct role in inhibiting these transcription factors which may lead to positive\negative feed back on the Scl27a1 and Prdm6. This needs to be discussed by siting previously published work of direct effects of RSV and NR on transcription factors.

Information regarding transcription factors (TF) that may modulate Slc27a1 and Prdm16 gene transcription in adipose tissue is given and discussed in parallel with the description of the putative TF binding sites mapped by MatInspector software in the Slc27a1- and Prdm16-related genomic regions analyzed, as we think this approach helps maintaining the work focus. In particular, information (including references) of possibly modulatory TFs affecting transcription is included in the last paragraph of section 3.1 for Slc27a1, the last paragraph of section 3.2 for Prdm16, and a paragraph in the Discussion (between lines 486-493) for joint discussion.

Concerning the possibility of common regulation of Scl27a1 and Prdm16, interestingly regulatory binding sites for nuclear respiratory factor 1 (NRF1) were identified for both genes through MatInspector. These sites and their possible functional relevance are highlighted in the Discussion, lines 488-493, were we write: “In particular, putative binding sites for NRF1 - a TF related to mitochondria biogenesis and dynamics [48] - were identified in the Slc27a1 distal promoter and Prdm16 proximal promoter regions analyzed, close to CpGs that were found to be hypomethylated in the treated animals, under NFD or both NFD and HFD. Importantly, Slc27a1 and Prdm16 are both putative target genes of NRF1 according to ChIP-seq results in the ENCODE Transcription Factor Targets data set [49,50]”.  We agree with the Reviewer that, to round the picture, the manuscript will benefit from the inclusion of literature on the effect of RSV and NR on transcription factors modulating expression of studied genes. We now include this information in the Discussion, restricted to NRF1 to keep the work in focus. Thus, just after the sentence above, we have added the following sentence: “Moreover, RSV [51-53] and NR [54] both have been shown to induce NRF1 expression in experimental models as part of their stimulatory effects on mitochondrogenesis mediated by the activation of the SIRT1-PGC1a axis”. This has implied the addition of new references (52, 53 and 54).

Other changes introduced in the Ms:

- The title of Table 1 has been moved to the top.

- In Materials and Methods subsection 2.3. Tissue DNA extraction and bisulfite-sequencing PCR: the formula used to calculate the degree of C methylation is now better written in full.

- Figure 2 letters and numbers (other than the axis numbers) have been enlarged.

- A few typos have been corrected across the Ms.

- Small changes have been introduced in the Figure legends, to easy the reading and (for Figure 3) to include the description of the control (Co) group, which was missed in that legend by mistake.

- The term “cyclophilin A” previously used in the subsection of Methods dealing with “Cellular mitochondrial DNA content” has been substituted by the official name of this gene, Ppia, for the sake of coherence, since Ppia is the term used in previous lines in the text to refer to the cyclophilin A gene.

- references with[ppi] have been checked and the corresponding doi corrected (this affects to current references 2, 11, 14, 16, 18, 24, 51 and 57).
